# Association of Sleep Duration and Overweight/Obesity among Children in China

**DOI:** 10.3390/ijerph17061962

**Published:** 2020-03-17

**Authors:** Jing Fan, Caicui Ding, Weiyan Gong, Fan Yuan, Yan Zhang, Ganyu Feng, Chao Song, Ailing Liu

**Affiliations:** Department of Nutrition and Health Education, National Institute for Nutrition and Health, Chinese Center for Disease Control and prevention, Beijing 100050, China; jing_zwtcheroyl@163.com (J.F.); dingcc@ninh.chinacdc.cn (C.D.); gongwy@ninh.chinacdc.cn (W.G.); yuanfan@ninh.chinacdc.cn (F.Y.); zhangyan@ninh.chinacdc.cn (Y.Z.); fenggy@ninh.chinacdc.cn (G.F.); songchao@ninh.chinacdc.cn (C.S.)

**Keywords:** sleep duration, overweight, obesity, children, China

## Abstract

To investigate the association of sleep duration with overweight and obesity among children aged 6 to 17 years in China, 2010–2012 data from the China National Nutrition and Health Surveillance (CNHHS) were analyzed. A total of 35,414 children were recruited in the survey. Body mass index (BMI) was converted into three categories: normal weight, overweight and obesity. In multinomial logistic regression model, sleep duration was divided into four groups: very short, short, recommended and long. In restricted cubic splines (RCS), sleep duration was examined as a continuous variable in relation to overweight and obesity. In the very short and short groups, sleep duration was a risk factor for obesity after adjusting for the potential impacts of age, gender, residence, family income, leisure sedentary behavior (SB) and leisure exercise, with OR (Odds Ratio) = 3.01 (95% CI (confidence interval): 2.19–4.15) and OR = 1.24 (95% CI: 1.14–1.35), respectively. The adjusted OR of overweight for short sleep duration relative to a recommended sleep duration was 1.17(95% CI: 1.09–1.26). No significant associations of very short sleep with overweight, of long sleep duration with overweight and obesity were found. The RCS curves between sleep duration and overweight and obesity were both inverted J-shaped. To conclude, the shorter the sleep duration, the higher the risk of overweight and obesity in children. Increasing sleep duration would have a positive effect on reducing overweight and obesity rates in Chinese children.

## 1. Introduction

Globally, the prevalence of overweight and obesity among children aged 2 to 19 years has increased by 47.1 percent from 1980 to 2013. The dramatic rise trend over the past few decades is not confined to economically privileged countries (boys: from 16.9% to 23.8%, girls: from 16.2% to 22.6%), but also increasing rapidly in a growing number of developing countries (boys: from 8.1% to 12.9%, girls: from 8.4% to 13.4%) [1]. A survey on the physique and health of Chinese students showed that during the nearly 30 years from 1985 to 2014, the prevalence of overweight and obesity among Chinese students aged 7 to 18 years showed a trend of steady increase, from 1.4% to 12.1%, 0.1% to 7.3%, respectively [2].

Excessive body weight is often positively associated with diabetes, high blood pressure and other chronic diseases [3,4,5]. The occurrence of these diseases can impose a profound economic burden on the country. In 2010, the combined economic burden of major chronic diseases caused by overweight and obesity in urban and rural China reached 90.77 billion RMB, accounting for 42.9 percent of the economic burden of major chronic diseases [6]. The risk factors for overweight and obesity, such as high energy intake, low fruit and vegetable consumption, less physical activity, genetic factors, have always been researched hotspots in many published literatures [7,8,9]. Recently, attentions have been turned to inadequate sleep, an emerging influential factor [10,11]. Sleep is a complex brain process that sustains the body’s basic physiological needs [12]. Many countries around the world suffer from people of all ages getting less than the recommended amount of sleep [13]. In 2015, data from a Centers for Disease Control and Prevention(CDC) survey on risky behavior in children and adolescents in the United States showed that 57.8% of children in grades 6 to 8 and 72.2% of children in grades 9 to 12 did not get enough sleep while in school [14]. Sleep deprivation has been discussed in previous studies as a possible link to diseases such as obesity, cardiovascular disease, diabetes and malignant tumors [15,16] and experimental studies show that sleep deprivation could lead to weight gain in adults [17]. Sleep deprivation may also be present in Chinese children, while the previous studies on the association of sleep deprivation with overweight/obesity among Chinese children lacked national data [18].

The shape of the curve between sleep duration and overweight/obesity was inconsistent in different studies. In a cross-sectional study involving 66,817 subjects [18], Wu et al. suggested that when sleep duration was maintained at 7-8 hours, the impact on overweight and obesity was the lowest, and lower or higher than the optimal zone would increase the risk of overweight and obesity. The U-shaped relationship was also discussed in earlier epidemiological studies [19,20]. A meta-analysis [21] involving 56,584 children showed a moderate negative trend between sleep duration and the risk of overweight and obesity duration. For each additional hour of sleep per day, the risk of being overweight/obese was reduced by 21%. Recently, Zhou et al. [22] evaluated the dose-response relationship between sleep duration and overweight and obesity in a systematic review of prospective cohort studies. The curve was in an inverted J shape, and there was no significant ascending trend after sleep duration surpass 7 to 8 hours. In addition, some epidemiological studies have not found an association between sleep duration and overweight/obesity as well [18,23,24,25,26]. Few studies on the association of sleep status with overweight and obesity among nationally representative Chinese children are available. Therefore, the current study used the data from a national representative cross-sectional survey to explore the association of sleep duration with overweight and obesity among Chinese children.

## 2. Materials and Methods

### 2.1. Study Participants

The data were from the China National Nutrition and Health Surveillance (CNHHS) in 2010–2012. The multi-stage stratification and population proportional cluster random sampling method was adopted. All county-level administrative units were divided into four categories: big cities, small and medium-sized cities, ordinary rural areas and poor rural areas. A total of 150 counties (districts) were selected from four categories of areas as study sites. Then proportional sampling approach was adopted to extract six villages or communities at equal intervals from each county/district. Finally, 75 households were randomly selected from each village/community and children aged 6 to 17 years in each family were involved. If the number of children was less than 20 children (10 boys and 10 girls) in each age group in each site, supplementary children would be selected from nearby primary and secondary schools to meet the minimum sample size. Finally, a total of 38,744 children aged 6 to 17 years were investigated [27]. After excluding missing and abnormal data in height, weight, sleep duration and sedentary time, 35414 subjects were included in the study.

An interview-administered questionnaire was used to collect the information about sedentary duration, leisure exercise and sleep duration by trained investigators. The children younger than 10 years old finished the questionnaire with the help of their parents. This study was approved by the ethics review committee of the National Institute for Nutrition and Health, Chinese Center for Disease Control and Prevention (No. 2013-018), and all participants’ parents or legal guardian signed the informed consent.

### 2.2. Anthropometric Measurements

Height and weight were measured for all participants. All measurements were conducted by well-trained investigators under standard operation procedure. The height was measured with an accuracy of 0.1 cm and the weight was measured with an accuracy of 0.1 kg [27].

### 2.3. Definition of Overweight and Obesity

Overweight and obesity was classified according to age- and gender-specific BMI cutoff points which developed for Chinese children [28].

### 2.4. Categories of Sleep Duration

The sleep duration was divided into four levels according to the National sleep foundation (NSF) guideline for age-specific sleep recommendations [29]: very short, short, recommended and long sleep duration. For children aged 6 to 13 years, sleep duration < 7h/d was classified as very short, within 7 to 8 h/d as short, within 9 to 11 h/d as recommended, and more than 11 h/day as long. For children aged 14 to 17 years, sleep duration < 6h/d was classified as very short, within 6 to 7 h/d as short, within 8 to 10 h/d as recommended, and more than 10h/d as long.

### 2.5. Statistical Analyses

All data were analyzed using the Statistical Analysis System (SAS) 9.4 software (SAS Institute Inc., Cary, NC, USA). Mean and standard deviation (SD) were used to describe the continuous variables, and the classified variables were presented in the form of frequency and percentage. Differences in demographic characteristics and physical activity among the four sleep groups were examined using one-way analysis of variance (ANOVA) or Chi-square test. Multinomial logistic regression was applied to examine the association of sleep duration with overweight and obesity, where dependent variable was BMI status (normal weight, overweight and obesity), and independent variable was the level of sleep duration (recommended sleep group = 0, very short sleep group = 1, short sleep group = 2, and excessive sleep group = 3). Moreover, gender (male = 0, female = 1), age, residence (urban = 0, rural = 1), family income (low income = 0, middle income = 1, high income = 2), leisure exercise level (no leisure exercise = 0, leisure exercise = 1) and leisure sedentary time level (not more than 2 h/d = 0, more than 2 h/d = 1) [30] were included in the model as confounders. Additionally, in order to obtain a more objective and realistic correlation in this study, we took sleep duration as a continuous variable and plotted its nonlinear relationship with overweight and obesity using restrictive cubic splines (RCS).

## 3. Results

### 3.1. The Distribution of the Characteristics among Different Sleep-Duration Groups.

The overall proportion of very short, short, recommended, long sleep duration was 0.1%, 36.1%, 62.6% and 0.1%, respectively, among Chinese children aged 6–17 years. Compared with their counterparts, boys, rural residents, those with low family income, with sedentary duration > 2h/d and no leisure exercise had higher proportion of recommended sleep duration. Sleep deprivation was more likely to occur in older children. Among the four sleep groups, significant difference in the rates of overweight and obesity was found, and highest in the very short group (see Table 1).

### 3.2. Multivariate Regression Analysis of the Relationship between Sleep Duration and Overweight/Obesity.

As shown in Table 2, compared with children with recommended sleep duration, those with very short sleep duration and short sleep duration had higher risk of obesity, with adjusted OR (odds ratio) = 3.01 (95% CI (confidence interval): 2.19–4.15) and adjusted OR = 1.24 (95% CI: 1.14–1.35), respectively. 

Those with short sleep duration had a higher risk of overweight (adjusted ORs = 1.17, 95% CI: 1.09–1.26). No significant relationship was found between long sleep duration and overweight and obesity. In addition, the risk of overweight and obesity was higher among younger children, boys, urban children, those with sedentary duration > 2h/d, those doing leisure exercise, and those with higher family income.

### 3.3. Restrictive Cubic Splines Demonstrate the Relationship between Sleep Duration and the Risk of Overweight/Obesity.

The quantiles of cumulative sleep time (25th, 50th, and 75th percentiles) were selected as knots for plotting the RCS curve, fitting the spline function piecewise and smoothing the connection at the break point, as shown in Figure 1. The corresponding ORs and 95% CI of the duration of sleep at the integral point in the curve with 8.5 h/d as reference were listed in Table 3. The number of children sleeping more than 12.0 h/d was too small, so the corresponding ORs were not listed. Overall, both the curve for overweight and obesity were inverted J shaped trend. With the increase of sleep duration, the risks of overweight and obesity decreased. Compared with children with 8.5 h/d sleep, children with 4.0/d sleep had a higher risk for overweight (OR = 1.88, 95% CI: 1.40–2.54) and obesity (OR = 2.55, 95% CI: 1.78–3.65). 

## 4. Discussion

The current study explored that Chinese children who slept less than the recommended range were more likely to be overweight and obese by both RCS and multivariate logistic analysis. With the increase of sleep duration, the risk of overweight and obesity gradually decreased. In terms of how sleep deprivation affects weight, Data et al. [31] found the positive relationships between sleep deprivation and blood glucose fluctuations, negative mood, and cravings in an n-of-1 randomized pilot study. The association of sleep deprivation with blood glucose fluctuations also showed a similar trend in a review [32]. These consequences contribute to high calorie intake, which leads to obesity. Moreover, sleep deprivation may result in metabolic disorders like abnormal leptin levels, which can also lead to obesity [33].

The results of the current study are consistent with those of previous studies [34,35,36], which showed an inverse relationship between the risk of overweight and obesity and sleep duration. While the trend of RCS curves in this study is not consistent with h conclusions of Wu et al. [18]. They recruited 66,817 school-age children in Guangzhou Province and plotted cubic splines based on their personal characteristics and sleep duration. The risk of being overweight and obese was lowest within a certain period (7–8 h/d), and both insufficient sleep and excessive sleep outside that period increase the risk. This U-shaped curve relationship has also been revealed in another studies [37]. However, in our study, we did not observe an increase trend in overweight/obesity rates associated with excessive sleep duration. This may be due to the different distribution of participants in different sleep groups. The number of children in our study who got more than the recommended amount of sleep was extremely small. Therefore, the relationship between excess sleep duration and overweight and obesity may be influenced. In fact, a negative relationship between sleep duration and BMI was found in a number of pediatric cross-sectional and longitudinal studies [38,39,40,41,42], and U-shaped curves were often pointed out in adults [43,44]. Marshall et al. argued that because of the ceiling effect, few children sleep excessively long in their daily lives [45]. Currently, there are many studies on the relationship between sleep restriction and overweight and obesity in adults and children, but the effect of excessive sleep duration on body weight is not well understood. The information from Chinese population will be helpful. In our study, all respondents were recruited from the CNHHS, a survey covered 31 provinces, autonomous regions and province-level municipality in China, which can provide more information for the understanding of this area. In our study, consistent with previous findings, girls, older children, and rural children had lower risk of overweight and obesity [30,31,32]. Children doing leisure exercise were found to have a higher rate of overweight and obesity than those without leisure exercise in the current study. This may be related to the fact that most Chinese children paid attention to their weight [34], and those who were dissatisfied with their weight were more likely to conduct weigh loss behaviors, including participating in exercise [46]. It is also possible that the children who did leisure exercise had higher muscle mass, which made them heavier and thus wrongly classified as overweight or obese.

Our study had some limitations that need to be addressed in any future work. Firstly, this national survey was cross-sectional, thus, the directionality of the causation cannot be established. Further prospective cohort studies are warranted if permitted. Secondly, energy intake was not taken into consideration in the study, and high energy-dense eating habits plays an important role in weight gain. In addition, the sleep duration in the study was self-reported, which may differ from the actual sleep duration.

## 5. Conclusions

The prevalence of overweight and obesity among Chinese children may be linked to inadequate sleep duration. With the increase of sleep duration, the risk of overweight and obesity gradually decreased, and the trend gradually leveled off. The overall curve presented an inverted J-shape. Ensuring adequate sleep duration for children is of great significance in controlling overweight and obesity in children and healthy lifestyle education should be conducted for children to contain the rapid increase of overweight and obesity for Chinese children.

## Figures and Tables

**Figure 1 ijerph-17-01962-f001:**
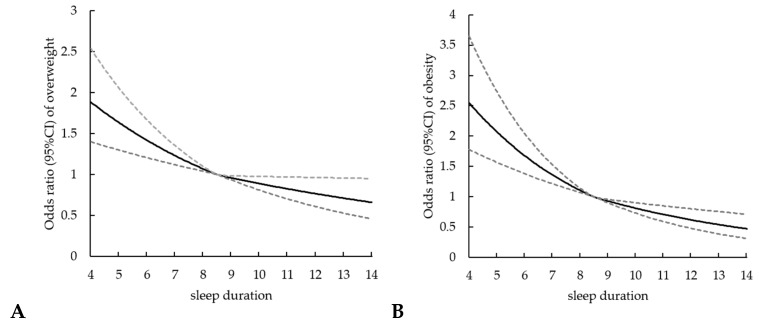
Non-linear regression curves of associations of sleep duration with overweight (**A**) and obesity (**B**).

**Table 1 ijerph-17-01962-t001:** Demographic characteristics and physical activity of different sleep groups.

Variables	Total	Very Short	Short	Recommended	Long	*p*
	*N*	*N* (%)	*N* (%)	*N* (%)	*N* (%)	
Total	35,414	308 (0.1)	12776 (36.1)	22158 (62.6)	172 (0.1)	
Gender						
Male	17,859	142 (46.1)	6334 (49.6)	11297 (51.0)	86 (50.0)	<0.0001
Female	17,555	166 (53.9)	6442 (50.4)	10861 (49.0)	86 (50.0)	
Age (years)	35,414	13.23 ± 3.2	12.31 ± 3.1	11.56 ± 3.4	11.66 ± 3.4	<0.0001
Region						
Urban	17,826	211 (68.5)	6589(51.6)	10940(49.4)	86(50.0)	<0.0001
Rural	17,588	97 (31.5)	6187(48.4)	11218(50.6)	86(50.0)	
Family income						
Low income	22,298	185 (60.1)	7940 (62.2)	14,065 (63.5)	108 (62.8)	<0.0001
Middle income	4267	49 (15.9)	1682 (13.2)	2520 (11.4)	16 (9.3)	
High income	1119	24 (7.8)	395 (3.1)	691 (3.1)	9 (5.2)	
Unknown	7730	50 (16.2)	2759 (21.6)	4882 (22.0)	39 (22.7)	
Leisure exercise						
No	22,088	160 (52.0)	8001 (62.6)	13,852 (62.5)	75 (43.6)	<0.0001
Yes	13,326	148 (48.1)	4775 (37.4)	8306 (37.5)	97 (56.4)	
Leisure SB						
≤2 h/d	14,241	68 (22.1)	4539 (35.5)	9546 (43.1)	88 (51.2)	<0.0001
>2 h/d	21,173	240 (77.9)	8237 (64.5)	12,612 (56.9)	84 (48.8)	
BMI status						
Obesity	2779	52 (16.9)	1028 (8.1)	1687 (7.6)	12 (7.0)	<0.0001
Overweight	3736	36 (11.7)	1443 (11.3)	2246 (10.1)	11 (6.4)	
Normal	28,899	220 (71.4)	10,305 (80.7)	18,225 (82.3)	149 (86.6)	

**Table 2 ijerph-17-01962-t002:** Multivariate logistic regression analysis of sleep duration and overweight and obesity.

Variables	Overweight	Obesity
	Odds Ratio (95% CI (Confidence Interval))	OR (95% CI)
Sleep duration		
Very short	1.34 (0.94–3.01)	3.01 (2.19–4.15)
Short	1.17 (1.09–1.26)	1.24 (1.14–1.35)
Recommended	reference	reference
Long	0.58 (0.32–1.08)	0.83 (0.46–1.51)
Gender		
Male	reference	reference
Female	0.69 (0.64–0.73)	0.57 (0.52–0.61)
Age	0.95 (0.94–0.96)	0.85 (0.84–0.86)
Residence		
Urban	reference	reference
Rural	0.69 (0.64–0.74)	0.62 (0.57–0.68)
Family income		
Low income	reference	reference
Middle income	1.22 (1.10–1.35)	1.30 (1.16–1.46)
High income	1.20 (1.20–1.44)	1.29 (1.06–1.58)
Leisure exercise		
No	reference	reference
Yes	1.16 (1.08–1.25)	1.27 (1.17–1.38)
Leisure sedentary behavior		
≤2 h/d	reference	reference
> 2h/d	1.10 (1.02–1.18)	1.10 (1.01–1.19)

**Table 3 ijerph-17-01962-t003:** OR and 95% CI for the restrictive cubic splines RCS.

Sleep Duration (h/d)	Overweight	Obesity
4.0	1.88 (1.40–2.54)	2.55 (1.78–3.65)
5.0	1.63 (1.30–2.06)	2.07 (1.57–2.73)
6.0	1.42 (1.21–1.67)	1.68 (1.38–2.04)
7.0	1.23 (1.12–1.35)	1.36 (1.21–1.52)
8.0	1.07 (1.04–1.10)	1.10 (1.07–1.14)
9.0	0.96 (0.93–0.98)	0.93 (0.90–0.96)
10.0	0.89 (0.81–0.98)	0.81 (0.73–0.90)
11.0	0.82 (0.70–0.97)	0.71 (0.59–0.85)
12.0	0.77 (0.61–0.96)	0.62 (0.48–0.80)

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
