# Peer review of "Association of Sleep Duration and Overweight/Obesity among Children in China"

_ijerph, 2020, doi:10.3390/ijerph17061962_

Round 1
Reviewer 1 Report
Fan et al carried out a population study in Chinese children to assess the association of sleep duration with incidences of obesity/being overweight. The results of the study is somewhat expected, but may be important for this specific population. However, the manuscript suffers from certain issues as detailed below -
- The subjects have significant variation in age. Did the authors corrected for age when performing the analysis?
- It is not clear from the manuscript how overweight/obesity is defined. Specifically, overweight and obesity are usually defined as two different categories. It looks like the authors merged the two categories. Why?
- The subjects who underwent physical exercise are also prone to be overweight/obese.This is somewhat counterintuitive. How do the authors explain that?
- did the authors take into account any diseases, specifically, sleep disorders in the participants?
Reviewer 2 Report
Association of sleep duration and overweight/obesity among children in China
This is a robust analysis of a rich dataset, and will make a contribution to the evidence base between sleep and adiposity in children. However, improvements are needed before this can be assessed appropriately.
- The English needs to be improved to be able to understand certain parts of the manuscript.
- It is not stated when this data was collected.
- On line 82, exclusion of data is reported as ‘controlling for’.
- As this was from a survey, was survey analysis used? This would include the strata and clusters etc
- Exercise and sedentary time were dichotomised? Please describe how this was done in the methods.
- Table 1 should present column percentages (not row percentages) so that comparison can be made between the groups. E.g. did the gender composition of the groups look different? We can’t judge this with the numbers presented as they are.
- Please use age as a continuous variable in all analyses – the dichotomisation is arbitrary at the moment and sleep and age have a strong relationship so it is important to handle it as a continous variable.
- Report percentages to 1dp - this is a personal preference so this change is not mandatory, but this level of precision is appropriate and will make the table easier to read.
- Line 147 the risk was reported as 1.76 times higher, but I can’t find this result in Table 3?
- Care should be taken to acknowledge that this is cross-sectional data and so bidirectionality and confounding could explain the results (e.g. those with higher BMI might sleep less, not that those who sleep less have higher BMI).
- On lines 191 & 192: this is a sample of children, who tend to be less concerned with their weight status, so this is is perhaps less relevant here. Indeed, it may be that those who participate in exercise are more likely to have increased lean mass, making them heavier, and then misclassified as overweight.
Reviewer 3 Report
- This national data is valuable. The present analysis seemed too simple. The authors could examine the moderation of gender, age, residence, leisure sedentary behavior and leisure exercise. The results can provide more knowledge to the field.
- I wonder why the authors grouped overweight and obesity together. Previous studies found that there were differences in psychopathology and daily activities between children and adolescents with overweight and obesity. The sample size was large enough in the present study to divide the participants for further analysis.
- Did the authors collect parental education and socioeconomic status? They may moderate the association between sleep duration and body weight status.
Round 2
Reviewer 1 Report
The author mostly addressed my concerns. Hence, the manuscript may be accepted in present form.
Author Response
Thank you very much for your comments and suggestions for this article.
Reviewer 2 Report
Thank you. The authors have adequately responded to all of my comments.
A few minor items remain:
- Age now needs to have units reported (e.g. years)
- How was leisure time exercise and SB assessed?
- In Table 3, does 4 hours, mean <4 hours or between 4 & 5 hours or median of 4 hours. Please be specific.
The manuscript needs extensive English editing.
Reviewer 3 Report
The authors have revised their manuscript based on the reviewer's suggestions. I would like to suggest the editors accepting it for publication.
Author Response

(The authors gave the same response as above.)
